# Specific Binding of Alzheimer’s Aβ Peptides to Extracellular Vesicles

**DOI:** 10.3390/ijms25073703

**Published:** 2024-03-26

**Authors:** Christina Coughlan, Jared Lindenberger, Jeffrey G. Jacot, Noah R. Johnson, Paige Anton, Shaun Bevers, Robb Welty, Michael W. Graner, Huntington Potter

**Affiliations:** 1University of Colorado Alzheimer’s and Cognition Center (CUACC), Linda Crnic Institute for Down Syndrome (LCI), Department of Neurology, University of Colorado Anschutz Medical Campus, 13001 E. 17th Pl, Aurora, CO 80045, USAhuntington.potter@cuanschutz.edu (H.P.); 2Structural Biology and Biophysics Core, University of Colorado Anschutz Medical Campus, Aurora, CO 80045, USArobb.welty@cuanschutz.edu (R.W.); 3Duke Human Vaccine Institute, Duke University, 2 Genome Ct., Durham, NC 27710, USA; 4Department of Bioengineering, University of Colorado Anschutz Medical Campus, 13001 E. 17th Pl, Aurora, CO 80045, USA; 5Department of Neurosurgery, University of Colorado Anschutz Medical Campus, 13001 E. 17th Pl, Aurora, CO 80045, USA

**Keywords:** extracellular vesicles (EVs), amyloid, isothermal titration calorimetry (ITC), atomic force microscopy (AFM)

## Abstract

Alzheimer’s disease (AD) is the fifth leading cause of death among adults aged 65 and older, yet the onset and progression of the disease is poorly understood. What is known is that the presence of amyloid, particularly polymerized Aβ42, defines when people are on the AD continuum. Interestingly, as AD progresses, less Aβ42 is detectable in the plasma, a phenomenon thought to result from Aβ becoming more aggregated in the brain and less Aβ42 and Aβ40 being transported from the brain to the plasma via the CSF. We propose that extracellular vesicles (EVs) play a role in this transport. EVs are found in bodily fluids such as blood, urine, and cerebrospinal fluid and carry diverse “cargos” of bioactive molecules (e.g., proteins, nucleic acids, lipids, metabolites) that dynamically reflect changes in the cells from which they are secreted. While Aβ42 and Aβ40 have been reported to be present in EVs, it is not known whether this interaction is specific for these peptides and thus whether amyloid-carrying EVs play a role in AD and/or serve as brain-specific biomarkers of the AD process. To determine if there is a specific interaction between Aβ and EVs, we used isothermal titration calorimetry (ITC) and discovered that Aβ42 and Aβ40 bind to EVs in a manner that is sequence specific, saturable, and endothermic. In addition, Aβ incubation with EVs overnight yielded larger amounts of bound Aβ peptide that was fibrillar in structure. These findings point to a specific amyloid–EV interaction, a potential role for EVs in the transport of amyloid from the brain to the blood, and a role for this amyloid pool in the AD process.

## 1. Introduction

Extracellular vesicles (EVs), a generic term used to refer to cell-secreted vesicles, encompasses both exosomes (30–150 nm diameter) and microvesicles (50–1000 nm) [1,2]. EVs carry cargo that includes nucleic acids, proteins, lipids, and metabolites with these contents thought to reflect the health and/or disease state of their cell(s) of origin [3]. After their release, EVs can be secreted, degraded, or help with the recycling or transfer of plasma membrane components to target cells. To enter cells, EVs can selectively bind to surface components or fuse with lipid bilayers, both routes that allow EVs to share their physiologically benefiting cargo and/or their pathology-promoting contents [4,5,6,7,8,9,10,11,12,13,14,15,16,17,18,19,20,21,22,23,24,25]. 

With respect to exosomes, their generation begins in the endosomal system with early endosomes maturing into late endosomes or forming multivesicular bodies (MVBs) [7,15,26,27,28]. MVBs contain intraluminal vesicles (ILVs) formed by endosomal membrane invaginations into the endosomal lumen, modulated by ESCRT complexes and Rab GTPases, as well as ESCRT-independent pathways [4,29,30]. These ILVs can be targeted to the lysosome for degradation, or secreted from the cell as exosomes, a process that occurs when MVBs fuse with the plasma membrane under both normal and pathological conditions. With respect to microvesicles, they are generated by the outward budding and fission of the plasma membrane followed by the release of vesicles into the extracellular space [1,2]. 

After secretion, EVs can travel in biofluids and produce effects both locally and remotely [8,9,11,21,22,23,24,25,31,32,33] with the cargo and composition of EVs thought to reflect the physiological and pathological state of the parent cell [3,29,34,35,36,37,38,39]. EVs can enter cells by selectively binding to surface components, or via fusing with lipid bilayers [4,5,6,7,20,25,29,31,32,40]. Both routes of entry allow EVs to share their cargo locally and remotely, pointing to their potential role in both physiological and pathological processes. Thus, deciphering cargo that is selectively loaded into/onto EVs under normal physiological and pathological conditions is imperative. 

The presence of amyloid in the brain is central to defining whether individuals are on the AD continuum [41,42,43,44,45,46,47,48,49,50]. While EVs have been reported to carry Alzheimer’s Aβ peptides [10,11,14,15,17,18,21,23,33,51,52,53,54,55] little is known about the specificity of this loading and thus whether it is of physiological consequence. 

In this work, to determine if Aβ42 and Aβ40 amyloid peptides specifically bind to EVs, we first characterized the plasma isolated vesicles to confirm that they were EVs. Upon confirmation that we had isolated EVs from plasma we incubated them with Aβ42 or Aβ40 and subsequently analyzed them using single molecule array (SIMOA^®^). To determine the molecular specificity and thermodynamic properties of the interaction between Aβ peptides and EVs, we analyzed the binding of Aβ peptides and peptides with the same amino acid composition but a random sequence via isothermal titration calorimetry (ITC) and SIMOA. The use of ITC in a titration-based approach revealed saturable binding of Aβ40 or Aβ42 with EVs and SIMOA measures confirmed that folds more Aβ were present on/in the EVs post incubation with these peptides. The results point to Aβ having a more specific interaction with EVs than previously understood indicating a physiological role for the interaction of Aβ and EVs under both normal and pathological conditions such as AD, and the potential utility of this EV-associated amyloid to serve as an AD biomarker.

## 2. Results

Extracellular vesicles (EVs) derived from plasma have been reported to carry Alzheimer’s Aβ peptides [10,11,14,15,18,23,33,46,53,54,55], yet the details of this association have not been studied. In this work, we investigated this association in three ways: a SIMOA-based binding assay, ITC, and AFM. 

First, using MISEV guidelines [56], we wanted to ensure that what we were isolating and studying in this work were EVs. To this end, we measured their size (NTA, EM, AFM), markers known to be expressed on exosome subsets of EVs (EXORAY200B-4, Exoview), and tetraspanin expression for exosome EVs (Exoview) (Figure 1). Examining the expression of certain proteins (tetraspanins) as well as size allows us to more definitively say that we are working with EVs and not just, for example, pure lipoproteins or cell debris. Nanosight analysis (Figure 1a), which measures hydrodynamic radii and concentrations of EVs, indicated that the differential diameters for the materials acquired by precipitation were consistent with exosome-like small EVs (30–150 nm vesicles) and microvesicles (50–400 nm). (Figure 1b) Exo-Check Exosome Antibody arrays (Systems Biosciences EXORAY200B-4) have 12 total printed antibody spots, 8 of these are for known exosome markers (CD63, CD81, TSG101, Alix, FLOT1, EpCAM, ICAM1, and ANXA5). The other four spots on the blot comprised a GM130 spot to test for any contaminating cis-Golgi, with a positive control at either end of the blot to check that the HRP detection is working, and a blank spot which should yield no signal. As shown (Figure 1b) based on all these measures, our EV preparations contain exosomes. Transmission electron microscopy (Figure 1c), a dehydrating technique, indicated the primary presence of smaller EVs (~30–50 nm) and Exoview analysis in Figure 1d in which we measured CD63, CD81, and CD9 tetraspanins corroborates that we have exosomes as a component of our EV population, given they have tetraspanin surface markers. Given the presence of exosomes (30–150 nm) but also smaller amounts of vesicles of larger sizes (>150 nm) most likely microvesicles, we chose to use the term “Extracellular Vesicles” to capture all forms of EVs isolated and used in this work.

To determine binding of EVs to Aβ42 or Aβ40 we used a SIMOA-based analysis (Figure 2). EVs derived from control human plasma samples (controls) (n = 5) were found to have on average 0.4, 5, and 1.2 pg/mL of Aβ42, Aβ40, and Tau, respectively (Figure 2A). Overnight incubation of control human plasma-derived EVs (n = 3), with a final concentration of monomeric 0.015 μM Aβ42, Aβ40, or scrambled peptides and post-incubation SIMOA analysis displayed elevated levels of peptide: 142.68 pg/mL and 550.73 pg/mL for Aβ42 and Aβ40, respectively. The observation that little to no Aβ40 (0 pg/mL) or Aβ42 (1.1 pg/mL) were detected when scrambled versions of the peptides were incubated with the EVs implies that these peptides do not act as mimics for A**β**40 or A**β**42 (Figure 2B,C). Performing the EVs precipitation isolation steps with Aβ alone (no EVs) did not provide any measurable Aβ.

The specificity of the EV–Aβ interaction was studied using isothermal titration calorimetry (ITC) (Figure 3). For the ITC experiments, after optimization it was decided to place the Aβ solution in the calorimeter and to inject in the extracellular vesicles. This injection process was initiated with 0.4 μL of extracellular vesicles, followed by 19 subsequent injections of extracellular vesicles of 2 μL per injection. Saturable binding was reached using this titration-based approach and the concentration of extracellular vesicles in each injected volume is described and summarized (Table 1). In summary, a total of 38.4 μL of EVs were injected and the concentration of EVs injected were calculated based on the starting EV concentrations (NTA). For example, for the EV preparation used in the ITC experiments described in Figure 3 the exosome concentration was 0.231 × 10^12^ particles per mL (Table 1). Given that a total volume of 38.4 μL of extracellular vesicles were injected into the calorimeter, this would equal a total of 0.887 × 10^10^ extracellular vesicles being injected over the course of the ITC experiment, 0.092 × 10^9^ at the first injection and then 0.46 × 10^9^ at each subsequent injection (a total of 19 of these injections), for a total EVs in the ITC experiment of 0.887 × 10^10^. This titration of EVs into Aβ peptides exhibited an endothermic isotherm, while the titration of EVs into either buffer (control titration) or scrambled peptides (Aβscr) exhibited negligible exothermic isotherms. The endothermic nature of the binding observed between the Aβ peptides and the EVs likely reflects the energy needed to overcome the loss of entropy for the Aβ that occurs when amyloid is organized onto EVs, including in structures such as fibrils. AFM imaging of Aβ42 confirmed that fibrillar forms are present in solution when Aβ42 was incubated with plasma extracellular vesicles overnight (Figure 4) while mainly non-fibrillar forms were observed when samples of peptides were examined subsequent to the ITC experiments. Since saturation occurred after short periods of interaction (30–60 min), it was not surprising to see less mature fibril forms (Figure 5). Given the diameter differences between the EVs that comprised exosomes (30–150 nm) and microvesicles (50–1000 nm possible size, we observed 50–450 nm) and Aβ (~1.4 pm), examining both simultaneously using AFM is not possible, (unless Aβ peptides form larger aggregates, which were sometimes visible). Given that saturable binding between extracellular vesicles and Aβ40 or Aβ42 in the ITC analysis occurred at low nM concentrations of Aβ the predominance of non-fibrillar Aβ forms is also not surprising. While Aβ40 bound to EVs in a predominantly endothermic manner, Aβ42 showed variability in the profiles, but when fibrillar Aβ42 was generated in vitro and used as the form of Aβ42 in the assay, the profile was consistently endothermic. This implies that there is a preference for binding fibrillar Aβ42 over the more monomeric forms, whereas all forms of Aβ40 (monomeric, oligomeric, and fibrillar) bind in an endothermic manner. The scrambled forms of Aβ40 and Aβ42 did not show measurable binding from the perspective that they did not show measurable amounts of exothermic or endothermic profiles.

## 3. Discussion

Many elegant studies [10,11,14,15,18,23,33,46,53,54,55], using techniques that include Western blotting, transcriptomics, proteomics, ELISA, PCR, immunohistochemistry, immunocytochemistry, and electron microscopy, have been directed at the understanding of the role of extracellular vesicles in the pathology of AD and in particular at the significance of Aβ-EV associations. While there is agreement that exosomes carry Aβ, there is a wide range of views on whether the purpose of this association is physiologically beneficial or pathological, the conclusions of which are also dependent on whether the authors of these works consider Aβ in its fibrillar form to be more toxic or protective [59,60,61,62,63,64,65,66,67,68,69,70,71,72,73,74,75,76]. Research points to many potentials. For example, the monomeric/oligomeric forms of Aβ can potentially be considered more toxic, due to their mobility and ability to be secreted and cross membranes in a less impeded manner; meanwhile, others show that the fibrillar forms are more toxic and need to be cleared [77,78,79,80,81,82]. Thus, are the exosomes gathering Aβ and clearing it or spreading it pathologically? This is an area of much interest.

To determine the significance of Aβ being present in or on EVs we must first understand if this interaction is specific. If the presence of Aβ merely reflects the binding of the hydrophobic corona on EVs to another hydrophobic partner (amyloid), then this interaction would be variable and of no utility as a biomarker of disease nor as a consistent cellular tool to spread or clear pathology. To this end, these studies are focused on measuring the specificity of the interaction between Aβ and EVs. This lays the groundwork for a reason to be interested in how Aβ binds to exosomes, uncovering whether Aβ is present in or on extracellular vesicles, dissecting the role of Aβ-laden extracellular vesicles in the pathology of disease or physiology, the role(s) of surface molecules in the binding of various forms of Aβ, deciphering the biological mechanisms involved in loading Aβ into/onto EVs, and the potential reasons for the proximity of EVs to plaques. In addition, there is much interest as to whether EVs clear or spread toxic peptides and pathology. If EVs bind Aβ in a specific manner, is it to reduce the formation of fibrillar forms? Do EVs bind Aβ monomers and lead to reductions in the time it takes to form fibrils and/or do they have a preference for one form of Aβ over another? This assembly of fibrillar Aβ and/or its preferential binding to EVs could point to the possibility that EVs may seed pathology using this attached Aβ/amyloid and/or clear it from the cell/body. 

The ITC experiments we have performed in this work point to this being a selective and saturable interaction that is energy requiring and hence possibly deliberate and important for cellular physiology and/or pathology. Aβ peptide binding to EVs detected using ITC unveiled that, while Aβ40 binding was endothermic and saturable, the more aggregation prone Aβ42 [83,84,85], was peptide-form dependent, with endothermic heat exchanges more robustly observed when preformed fibrillar Aβ42 (Aβ42F) was added to the assay. Thus, our ITC experiments suggest a preference for binding to fibrillar vs. monomeric forms of Aβ42. Questions still abound, e.g., when this Aβ–EV binding occurs, is it while the EVs are still in the cell (i.e., before the EVs are released into the extracellular space) or after they are released? Since EVs can easily traverse membranes, enter cells, and cross the blood–brain barrier, there are many interesting and important avenues of investigation that are ripe for investigation including the use of EVs for drug delivery in a less immunogenic manner. 

As previously mentioned, while there is more agreement that EVs carry Alzheimer’s Aβ peptides, to our knowledge fewer have investigated the specificity of this association; we deem this to be a crucial and important understanding if our goal is to establish a physiological or pathological relevance of loading Aβ into/onto EVs. The selectivity of the Aβ-exosome association points to the potential of using Aβ levels on EVs as dynamic and useful biomarkers for monitoring the progression of disease. Our finding that amyloid endothermically associates with EVs highlights the potential biological importance of this amyloid-EV pool that may affect the lipid organization of EVs or the proteins on their surfaces [86,87]. 

Thus, the ITC points to fibrillar forms being capable of interacting with and binding to EVs (maybe via the corona [88]/lipid bilayer) in a selective and saturable manner that requires energy. Given the energy input required if this is occurring in a cell/in the body, this would suggest an important role for this association, as the body does not use energy in a wasteful manner. Comparing the surface area of the exosomes against the concentration of Aβ, if there are ~500k Aβ per EV based on EV molarity and amount of Aβ added this would lead to an Aβ monomer binding every three angstroms. Given this is probably not physically possible, the most reasonable explanation is that Aβ oligomerizes before binding and that this interaction is selective and important.

Given that the form of Aβ42 affected the results points to a potential role for EVs in clearance and/or spreading of the aggregated forms of Aβ. While all peptide aliquots used were flash frozen post resuspension and stored at −80 °C, and that, according to AFM (Figure 5) and ThioT assay (Appendix A), the peptides utilized for ITC appeared monomeric or at most oligomeric, any variability in the isotherms could be explained by slight differences in the aggregation status of the peptide used for each assay. Given the greater propensity of Aβ42 to aggregate compared to Aβ40, the presence of small amounts of fibrillar Aβ42 in some of the aliquots could lead to the seeding of aggregation and more Aβ42F, leading to differences that are reflected as more robustly endothermic profiles. Scrambled forms of Aβ40 or Aβ42 showed negligible binding to EVs, as measured using the ITC and determined by the lack of measurable endothermic/exothermic profiles.

While AFM has many strengths and uses, in this work we used it to assess the starting forms of peptides used for these experiments and the forms that resulted following the incubation of the Aβ peptides with EVs. The fact that overnight incubations of EVs with Aβ showed that it was present in EVs and that longer incubations resulted in more fibrillar forms confirmed the aggregation capabilities of Aβ but also imply the possibility that EVs form a scaffold upon which the formation of fibrillar forms may occur. This assembly of fibrillar Aβ/or its preferential binding when presented to EVs can then point to the possibility that EVs may seed pathology using this attached Aβ/amyloid and/or clear it from the cell/body. In addition, since EVs can easily cross lipid membranes and easily enter cells, if Aβ/amyloid is selectively and stably attached we could assume that this Aβ/amyloid could then also be brought into a cell in this manner.

While we noted measurable saturable binding (n = 4), exact thermodynamic parameters of binding (ΔH, ΔG, ΔS, N) cannot be determined by this binding methodology due to a lack of exact knowledge of how many Aβs can bind to an EV. However, because saturable binding was observed [11,14,33,53,54,55], this points to the process of amyloid associating with EVs as being more specific than had previously been considered [10,15,17,18,21,23,51,52] and points to a physiological relevance of EVs in AD and likely to one or more specific Aβ-binding proteins on the surface of the EVs.

In summary, this work reveals that the interaction between amyloid and EVs is specific and saturable, which has biological implications. If EVs are involved in clearing amyloid, and EVs can only bind a finite amount of Aβ, under AD pathological conditions more EVs may be needed to bind to and remove the increased amounts of Aβ present. In addition, because clearing fibrillar amyloid to reverse pathology is an important focus of many therapeutics, these findings point to the need to better understand the role(s) of EV-associated amyloid as a possible target for interventions focused on improving Aβ clearance. The finding that, compared to monomeric Aβ42, Aβ42F preferentially associates with extracellular vesicles, suggests that if the endothermic type binding of Aβ42F does occur in vivo, this association and a possible role for EVs may be important for the clearance and/or spread of the fibrillar forms of Aβ42.

To our knowledge, this study presents a novel report in measuring the specificity of the Aβ–EV interaction, and points to this pool of Aβ in plasma as meriting consideration in normal and pathophysiological conditions, as a biomarker of physiological health or pathology, and/or as a read-out for changes in response to therapeutic interventions. Monitoring the trajectory of AD in a less invasive and more dynamic manner and/or determining the response to therapeutics utilizing a plasma biomarker is the focus of many in the AD field. The plasma pool of Aβ–EVs may be an important target for the development of both diagnostics and therapeutics.

## 4. Materials and Methods 

***Peptides:*** Recombinant human Aβ40 and Aβ42 (NaOH salts), and sequence scrambled versions, were purchased from rPeptide (Watkinsville, GA, USA). Peptides were reconstituted in Dulbecco’s phosphate-buffered saline (D-PBS, pH 7.4) (to maintain the peptide at a physiologically relevant pH, osmolarity, and ion concentration) at 200 μM concentration and sonicated for 10 min. For Aβ fibril formation, Aβ40 or Aβ42 were diluted to 20 μM in D-PBS, pH 7.4, and incubated at 37 °C with rapid shaking for 24 h. All Aβ preparations were flash frozen in liquid nitrogen and stored at −80 °C.

***Isolation of Extracellular vesicles:*** Extracellular vesicles were isolated from human plasma from healthy donors (control) by precipitation using EXOQ20a-1 precipitation reagent (Systems Biosciences LLC, 2438 Embarcadero Way, Palo Alto, CA 94303, USA). In summary, per 250 μl of plasma (purchased from BioIVT, Westbury, NY, USA), 67 μL of EXOQ20a-1 was added, mixed by inversion, and incubated on ice for 30 min. Subsequent to this incubation the solution was centrifuged for 10 min @3000 rpms and the resultant pellet resuspended in D-PBS at the same volume as that of the starting plasma used to isolate the EVs; in this example, this was 250 μL of D-PBS.

***Characterization of the Isolated Extracellular vesicles:*** Nanosight (NTA) (NS300 Malvern Analytics) analysis was used to assess the size and concentration of the vesicles [57] isolated by precipitation. This revealed that the EVs were in the size range expected for exosomes (30–150 nm) and microvesicles (50–450 nm). Due to the presence of EVs that were larger than exosomes (as determined using NTA), we have chosen to use the term “extracellular vesicles (EVs)” to include all particles isolated and studied in this work. 

***The Exo-Check Exosome Antibody arrays (Systems Biosciences, EXORAY200B-4):*** Initially the intention of this work was to focus on exosomes alone. However, with a precipitation-based approach for isolating exosomes, necessary for the small volumes of plasma available in cohorts and ideal for the development of a biomarker, the authors acknowledge the presence of and measured, using NTA, small numbers of other types of extracellular vesicles (EVs). For this reason, the authors wanted to be inclusive in the language used to describe the EV preparation. The Exo-Check Exosome Antibody arrays (Systems Biosciences LLC, 2438 Embarcadero Way, Palo Alto, CA 94303, USA); EXORAY200B-4) have 12 total printed antibody spots, 8 of which are for known exosome markers (CD63, CD81, TSG101, Alix, FLOT1, EpCAM, ICAM1, and ANXA5). These were used to display that the precipitated extracellular vesicles (EVs) contained exosomes. 

***Negative-staining Transmission Electron Microscopy***: was performed as previously described [58]. In summary, thin formvar/carbon film-coated 200 mesh copper EM grids were glow discharged for 1 min using a glow discharger. Purified EVs were fixed with 2% paraformaldehyde (PFA) for 5 min and 5–7 µL of the suspension solution (diluted 1:500 in D-PBS) was loaded onto the grid and incubated for 1 min. The EVs were immediately stained using the ~20 drops of filtered 1% uranyl acetate (UA) solution on the surface of the EM grid via syringe, and the excess UA solution was removed by contacting the grid edge with filter paper. The grid was then rinsed with a drop of water to remove excess staining solution and dried for 10 min at room temperature. Imaging was then performed via transmission electron microscopy (TEM) at 80 kV. 

***Exoview analysis:*** EVs were analyzed using Tetraspanin kits (Unchained labs: 251-1044, Pleasanton, CA, USA) and Exoview analysis was performed according to the manufacturer’s instructions. In summary, these chips capture exosomes based on the three most reported exosome tetraspanins and include a murine IgG control spot. The captured EVs can be subsequently stained for CD63, CD81, and CD9 or for markers of interest to determine, for example, how many CD81 captured extracellular vesicles have CD63 and CD9 on their surfaces. A sample image (Figure 1d) is provided displaying that our control human plasma EVs contain the most reported tetraspanins and thus are exosome-like EVs.

***SIMOA analysis:*** Aβ42 and Aβ40 peptide analysis was performed using the Quanterix Single Molecular Array (SIMOA^®^) SR-X Analyzer system, the Neurology 3-plex A (N3PA) kits and manufacturer’s instructions. Single molecular array (SIMOA) is an immunoassay technique that uses paramagnetic beads containing antibodies for the analytes of interest (e.g., Aβ42). These bead–antibody complexes are used to capture the analytes of interest from the sample. The captured analyte is then measured using a secondary antibody that is tagged with an enzyme that can cleave a fluorescent substrate and provide a signal that reflects the amount of captured analyte. The sensitivity of the assay arises from the ability of the technology to count each individual bead–analyte event for up to 240,000 events in single form, with even one captured analyte being sufficient to provide signal of sufficient intensity to be measurable. SIMOA has sensitivity in the femtomole range allowing us to measure the amounts of Aβ40 and Aβ42 that were bound to EVs in an accurate and sensitive manner. The scrambled versions were used to show that the sequence of the amino acids of Aβ is important for its binding, as no Aβ40 or Aβ42 signal could be detected when the amino acid sequence of Aβ40/Aβ42 is rearranged. While these standardized assays do not allow us to detect the scrambled versions themselves, we can use these findings to point to the lack of ability of the scrambled versions to act as mimics of Aβ40 or Aβ42.

***Binding of Aβ42 to extracellular vesicles***: (Figure 2). Preparations of EVs were generated from 250 μl of plasma from five human plasmas (purchased from BioIVT, Westbury, NY) from healthy donors (control) using EQULTRA-20A-1 precipitation reagent (Systems Biosciences) [57]. Each resultant EV pellet was resuspended in 250 μL of D-PBS; then diluted 1:2 (in D-PBS) and 125 μL of these resuspended EVs (corresponding to the equivalent of 62.5 μL of starting plasma) were pipetted into duplicate wells of a 96-well plate (USA Scientific Cat. No. 1833-9610). A total of 10 μL of either PBS, 0.2 μM monomeric Aβ42, Aβ40, Aβ42scr, or Aβ40scr was added to generate a final concentration (allowing for dilution in the assay) of 0 or 0.015μM Aβ, respectively. These Aβ concentrations were chosen based on optimization of the bound levels of Aβ that were within measurable ranges by SIMOA analysis [89,90]. The scrambled (scr) peptides contain all the amino acids found in Aβ40 and Aβ42 just arranged in a different sequence and do not form the amyloid structures normally observed for Aβ (confirmed by the R-peptide company and our AFM analysis). This mixture was incubated and shaken overnight at 37 °C. The EVs were then reprecipitated from this solution, lysed in 25 μL of 0.05 M Glycine and centrifuged @4000× *g* for 10 min. To the supernatant we added 3.75 μL of 1 M Tris pH 8.0, 6.25 μL of 3% BSA/PBS, and 45 μL mPERS with protease/phosphatase inhibitors. For complete lysis of the EVs, the resultant exosomal lysate was frozen (−80 °C) and thawed (37 °C) two times, and analyzed using the single molecule array (SIMOA^@^) Neuro-3-Plex assay as previously described [89,90]. 

***Isothermal Titration Calorimetry (ITC):*** Isothermal titration calorimetry (ITC) [91] experiments were performed at 25 °C using a Microcal ITC200 MicroCalorimeter (Malvern Panalytical, Westborough, MA, USA) in a ***titration-based*** format (Table 1). The calorimeter (cell) contained the peptides (20 μM in D-PBS) and the syringe contained the extracellular vesicles in D-PBS. In the optimization steps, 10, 20, and 25 μM concentrations of Aβ were tested, with 20 μM in D-PBS providing sufficient signal to measure heat exchanges. The approach of injecting extracellular vesicles into the Aβ solution was chosen due to concern that the ability of Aβ to aggregate and/or stick to syringes, etc., at varying degrees [92] might lead, upon injection, to the loss of some Aβ, in turn changing the concentration of Aβ in a manner that would not be accurately quantifiable. Measuring the heat exchanges of each of these concentrations when extracellular vesicles were added, along with considering the cost, led us to decide upon the use of 20 μM of the Aβ peptide in the calorimeter and injection of the EVs as being optimal. As shown in Table 1, we started with an injection of 0.4 μL and then made 19 subsequent injections of extracellular vesicles of 2 μL per injection. Saturation of binding was reached using this titration-based approach. The concentration of extracellular vesicles in each injected volume is summarized in Table 1, as is the overall final concentration of EVs used in each ITC experiment (total 38.4 μL injected) based on the starting EV concentrations (NTA). For example, for the EV preparation used in the ITC experiment (Figure 3) the concentration was 0.231 × 10^12^ particles per mL (Table 1). Given that a total volume of 38.4 μL of EVs was injected into the calorimeter, equaling a total of 0.887 × 10^10^ EVs being injected over the course of the ITC experiment, 0.092 × 10^9^ was injected at the first injection and then 0.46 × 10^9^ at each subsequent injection (a total of 19 of these injections) for a total EVs in the ITC experiment of 0.887 × 10^10^. The first injection of 0.4 μL was injected over 0.8 s and was excluded from data analysis. Subsequent injections were 2.0 μL over 4 s (19 total) with an injection spacing of 180 s. The reference power was set to 10 μcal/s in high feedback mode. The syringe stirring speed was set to 750 rpm. Data analysis was performed using Origin 7.0 software (OriginLab, Northampton, MA, USA).

***Sample preparation for Aβ42-exosome binding/Atomic Force Microscopy (AFM):*** EVs were generated from 250 μL of plasma using the methods previously described [57]. Each resultant EV pellet was resuspended in 250 μL of D-PBS to which either 10 μL of PBS or 20 μM Aβ42 was added for a final concentration of 0 or 0.8 μM Aβ42, respectively. This mixture was incubated with shaking overnight at 37 °C. EVs were reprecipitated and pelleted EVs were resuspended in Buffer B (Systems Biosciences LLC, 2438 Embarcadero Way, Palo Alto, CA 94303, USA) and analyzed using atomic force microscopy (AFM) versus the supernatant material.

***Atomic Force Microscopy (AFM):*** AFM images were acquired with a JPK NanoWizard 4a AFM (JPK Instruments USA, Carpinteria, CA, USA) with silicon-nitride cantilevers with a triangular tip and a nominal spring constant of 0.07 N/m (MLCT, Bruker AFM Probes, Camarillo, CA, USA) run in quantitative imaging (QI) mode. For sample preparation, 50 µL of sample was pipetted onto a clean, freshly cleaved mica surface and allowed to sit for 30 min. Samples were then washed twice with DI water. 

***ThioT Protocol:*** Thioflavin T is a benzothiazole salt that can bind to beta-sheet structures found in amyloid. When bound to beta sheets, ThioT emits a fluorescent signal measurable at approximately 482 nm, versus the unbound form which is measured at 427 nm. This alteration in signal is measured to determine the presence of amyloid structure using this assay. To assay the beta-sheet structure of our Aβ, 10 μL of the starting peptides was diluted by adding 40 μL of D-PBS (1:5 dilution). A total of 50 μL of starting peptide solution and 75 μL of ThioT mix (10 mL of 50 mM Glycine.PBS+27 μL of 5 mM ThioT.D-PBS) were added to this. The plate was incubated in the dark for 10 min and then excitation (440 nm) and emission (490 nm) were read using the bottom of the plate reader settings.

## 5. Conclusions

Taken together, these data show that extracellular vesicles bind Aβ peptides in a selective and specific manner, meriting consideration of this EV-associated amyloid in many roles, from clearance to catalysis of fibrillar Aβ formation. An association of Aβ with EVs could point to EVs binding to and catalyzing the formation of fibrillar forms (e.g., Aβ42F) intracellularly before the EV is secreted to the extracellular space. Alternatively, EVs may seed or catalyze the formation of fibrillar forms of Aβ in the extracellular space, post EV–Aβ secretion, due in part to their having some amyloid on their surface or due to their lipid membranes providing a scaffold for oligomerization. In addition, Aβ secreted on EVs may aid with clearance of Aβ and/or facilitate uptake of Aβ into naïve cells, possibly seeding aggregation and propagating disease. Regardless of its role, the association of Aβ with EVs points to Aβ levels in EVs as a useful biomarker for monitoring disease progression and response to therapeutic interventions. Further investigations into this utility are currently underway.

## Figures and Tables

**Figure 1 ijms-25-03703-f001:**
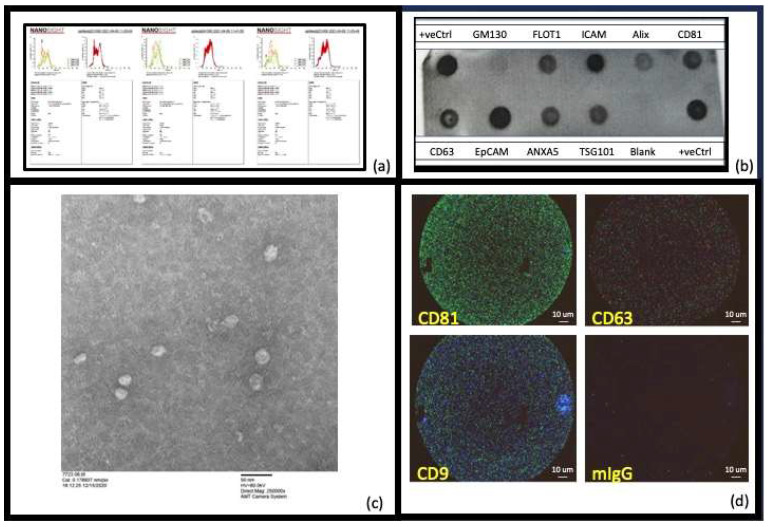
Characterization of the extracellular vesicles (EVs) isolated: (**a**) *Nanosight (NTA) (NS300 Malvern Analytics) analysis* was used to assess the size and concentration of the vesicles [57] and showed that the size of the EVs varied per participant, but all were in the range expected for EVs that comprised both exosomes (30–150 nm) and microvesicles (50–450 nm). Due to this participant-dependent variation in the amount of exosomes present, we have chosen to use the term “extracellular vesicles (EVs)” to include all particles isolated and studied in this work. (**b**) *The Exo-Check Exosome Antibody arrays (Systems Biosciences EXORAY200B-4)* have 12 total printed antibody spots, 8 of these are for known exosome markers (CD63, CD81, TSG101, Alix, FLOT1, EpCAM, ICAM1, and ANXA5). The other four comprised a GM130 spot to test for any contaminating cis-Golgi, with a positive control at either end of the blot to check the HRP detection is working, and a blank spot which should yield no signal. As can be seen, when our isolated EVs were incubated with these blots (after the proprietary EV labeling and detection was performed according to the manufacturer’s instructions), the exosome positive markers and the positive controls were strongly positive, the blank was negative and the negative control (GM130) had negligible signal. This confirmed that the EVs prepared using precipitation were extracellular vesicles that included exosomes. (**c**) *Negative-staining transmission electron microscopy,* a dehydrating technique summarized in Section 4 [58], indicated the primary presence of smaller EVs (~30–50 nm). (**d**) *Exoview analysis* allowed us to assess the presence of tetraspanins to determine if exosomes had been isolated (Unchained labs: 251-1044). The three most reported exosome tetraspanins, CD63, CD81, and CD9, were present in our healthy human control plasma EVS indicating the presence of exosomes.

**Figure 2 ijms-25-03703-f002:**
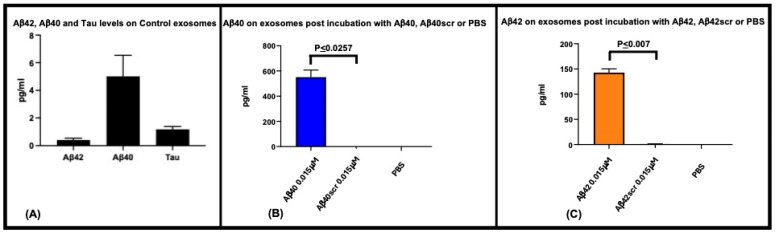
Aβ42 and Aβ40 bind to extracellular vesicles. To determine if A**β**42 and A**β**40 bind to extracellular vesicles, and to ensure the scrambled versions do not act as mimics we incubated 0.2 μM of each peptide (final concentration in assay; 0.015μM) with control plasma-derived EVs (from 62.5 μL of plasma) and measured the amount of Aβ42 or Aβ40 antibody detected peptide bound using the SIMOA 3-Plex assay (Quanterix). Control plasma EVs (n = 5) have on average 0.4, 5, and 1.2 pg/mL of A**β**42, A**β**40, and Tau, respectively (**A**). Using control plasma EVs (n = 3), and a starting incubation concentration of 0.015 μM A**β**42 and A**β**40, 142.68 pg/mL and 550.73 pg/mL of A**β**42 and A**β**40 were measurable, respectively. Incubation with scrambled versions of these peptides yielded minimal A**β**40 (0 pg/mL) and A**β**42 (1.1 pg/mL) signal (**B**,**C**) showing that in this assay of Aβ binding to exosomes, there is no background and that scrambled peptides do not appear to act as mimics of the native peptides.

**Figure 3 ijms-25-03703-f003:**
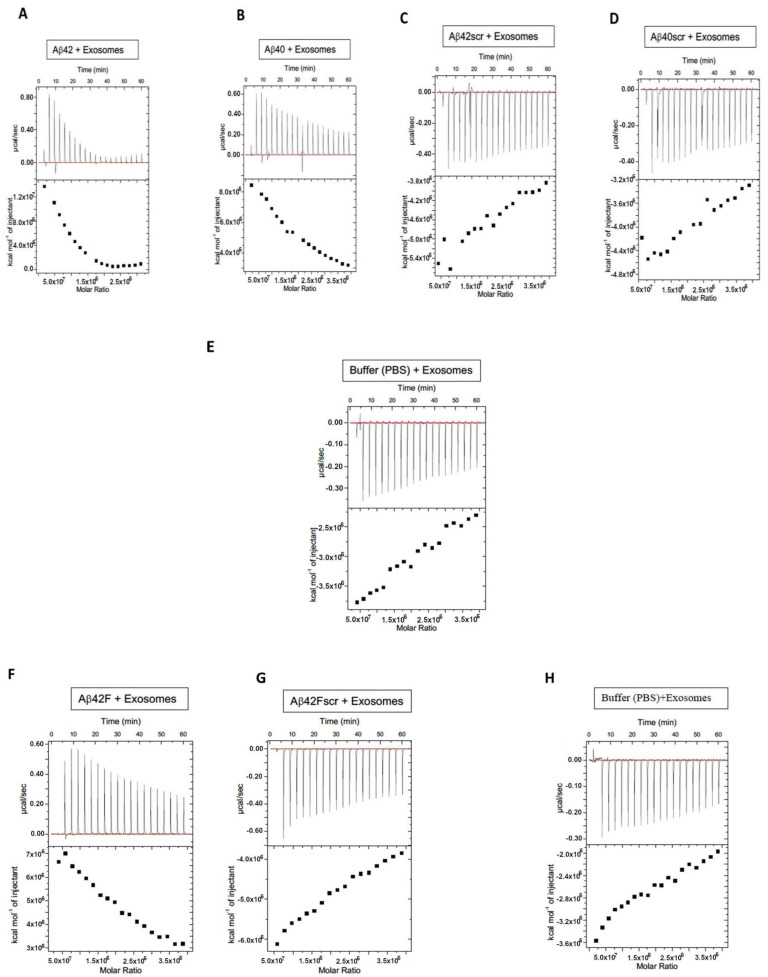
Aβ binds to extracellular vesicles in an endothermic and saturable manner. To assess the specificity of binding of Aβ40 and Aβ42 and their scrambled versions with extracellular vesicles (EVs), we utilized isothermal titration calorimetry (ITC). EVs were titrated into a solution containing one of the various types and forms of Aβ peptides: Aβ42 (**A**), Aβ40 (**B**), Aβ42scrambled (Aβ42scr) (**C**), Aβ40scrambled (Aβ40scr) (**D**), buffer alone (**E**), Aβ42Fibrillar (Aβ42F) (**F**), Aβ42Fibrillar scrambled (Aβ42Fscr) (**G**), and buffer alone for the fibrillar (**F**) assay (**H**). Appendix A shows repeats of these experiments, i.e., using control healthy plasma exosomes from three other subjects.

**Figure 4 ijms-25-03703-f004:**
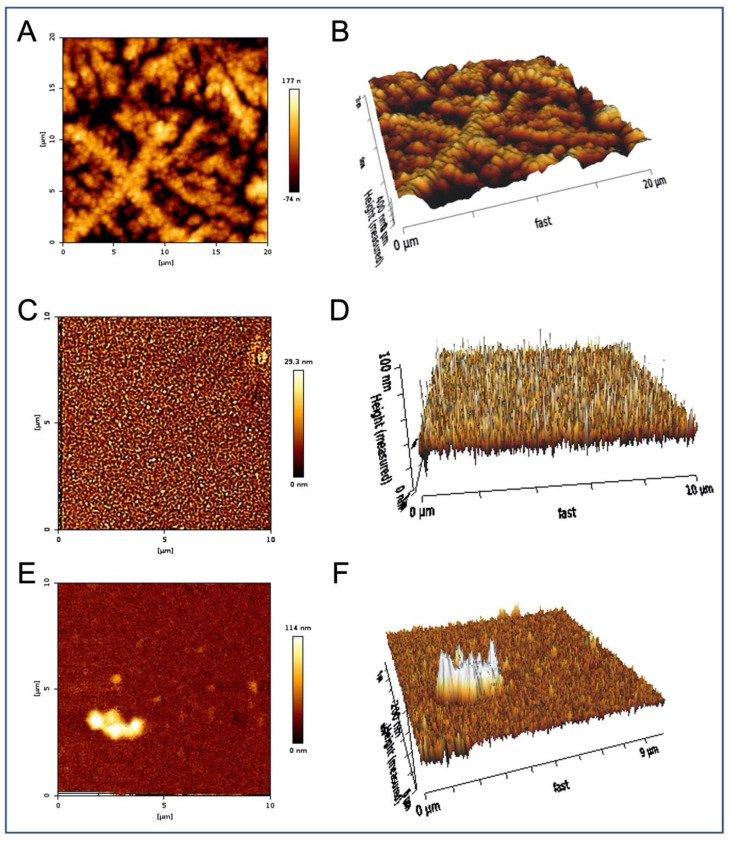
Extracellular vesicles incubated with Aβ42 overnight contain Aβ42 fibers. Representative AFM images. (**A**) EVs were incubated overnight with a final concentration of 0.8 μM A**β**42, then precipitated out of this solution, resuspended, and examined by AFM. The presence of long cylindrical A**β**42 fibers of around 80 nm diameter forming a 3D mesh was observed and measured. (**B**) 3D projection of (**A**). (**C**) Remaining supernatant (post precipitation of extracellular vesicles) shows individual particles of approximately 25 nm diameter, but no fibers. (**D**) 3D projection of (**C**). (**E**) Precipitation of EVs examined without A**β**42 incubation shows spherical structures around 100 nm diameter with indistinct (soft) surfaces. (**F**) 3D projection of (**E**).

**Figure 5 ijms-25-03703-f005:**
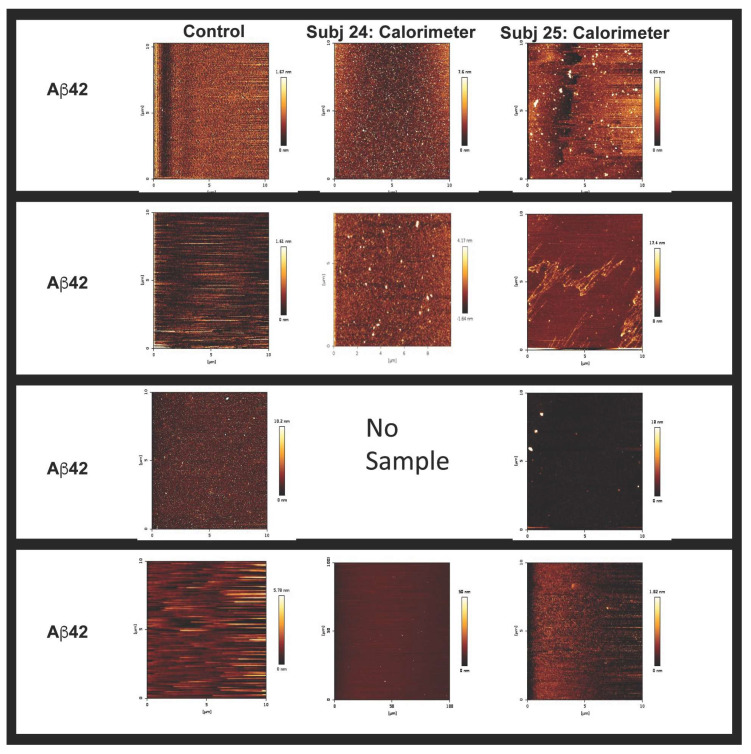
Calorimeter contents, post ITC, containing mainly non-fibrillar forms of Aβ. Examination of the calorimeter contents and peptides utilized for the assay by AFM displayed non-fibrillar forms of Aβ as being present in solution. Given that we did not have Subject 24’s Aβ40, we could not assess this sample using AFM.

**Table 1 ijms-25-03703-t001:** ITC assays were performed in a ***titration-based*** format.

Sample	Stock EVs (nM)	Concn EVs/mL	EVs First Injection (0.4 mL)	EVs Per Injection (19 × 2 mL)	Total EVs in ITC Assay (38.4 mL)
EVs Figure 2	3.85	0.231 × 10^12^	0.092 × 10^9^	0.46 × 10^9^	0.887 × 10^10^
Ppt Subj 23	18.9	1.14 × 10^12^	0.46 × 10^9^	2.3 × 10^9^	4.38 × 10^10^
Ppt Subj 24	18.3	1.1 × 10^12^	0.44 × 10^9^	2.2 × 10^9^	4.22 × 10^10^
Ppt Subj 25	21.1	1.27 × 10^12^	0.51 × 10^9^	2.55 × 10^9^	4.88 × 10^10^

## Data Availability

All data are contained within the article.

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
