# Peer review of "Specific Binding of Alzheimer’s Aβ Peptides to Extracellular Vesicles"

_ijms, 2024, doi:10.3390/ijms25073703_

Round 1

Reviewer 1 Report (New Reviewer)

Comments and Suggestions for Authors

Isothermal Titration Calorimetry (ITC) was used by the authors to reveal that the interaction between amyloid and extracellular vesicles (EVs) is specific and saturable, and the EVs are involved in clearing the amyloids. They discovered that Aβ42 and Aβ40, but not their scrambled control peptides, bind to the EVs in saturable and endothermic manners. I have the following comments.

The abstract id short and crisp. The Introduction is written well.

The materials and methods are described sufficiently, but needs further improvement. Describe in details the procedure of EVs isolation. Write down the full form of DPBS. Why DPBS was used to dissolve the peptides.

In the characterization of extracellular vesicles, The Exo-Check Exosome Antibody arrays part should be described a little more.

SIMOA analysis  should be described in more details.

When the Aβ 1-42 was added to the isolated EVS, what was its conformation? Was it a monomer, oligomer or matured fibrils? If it was fibrils, then what was the pretreatment done to induce fibrillation. Explain in the methodology section.

Thioflavin T assay was done and it is known that this dye specifically binds to the amyloid structure. How did the authors confirm that the Aβ 1-42 is in its fibrillar form?

 The discussion part needs to be improved.

Explain a little more about the scrambled peptides in the manuscript and how they are taken as non-amyloidogenic proteins.

General comments:

The English language needs to be corrected in the supplementary files of the manuscript. The figure captions of supplementary files are difficult to understand. Please make it lucid.

I recommend major revision.

Comments on the Quality of English Language

The langusge of the supplementary files needs to be improved

Author Response

REVIEWER 1:

Comments and Suggestions for Authors

Isothermal Titration Calorimetry (ITC) was used by the authors to reveal that the interaction between amyloid and extracellular vesicles (EVs) is specific and saturable, and the EVs are involved in clearing the amyloids. They discovered that Aβ42 and Aβ40, but not their scrambled control peptides, bind to the EVs in saturable and endothermic manners. I have the following comments.

The abstract is short and crisp. The Introduction is written well.

The authors thank the reviewer for this kind feedback.

The materials and methods are described sufficiently, but needs further improvement.

The authors thank the reviewer for this feedback and have added the additional details asked for below to this section to address the deficiencies shared by the reviewer.

Describe in details the procedure of EVs isolation.

This is now included in the Materials and Methods.

Write down the full form of DPBS. Why DPBS was used to dissolve the peptides.

The authors thank the reviewer for this feedback. The full form of the buffers name, Dulbecco’s Phosphate buffered saline (D-PBS), has now been shared in the manuscript. The main reason for using Dulbecco’s Phosphate buffered saline (D-PBS pH7.4) to resuspend the peptides was to maintain the peptide at a physiologically relevant pH, osmolarity and ion concentration.

In the characterization of extracellular vesicles, The Exo-Check Exosome Antibody arrays part should be described a little more.

The authors thank the reviewer for this comment.

Initially the intention of this work was to focus on exosomes alone. However, with a precipitation-based approach for isolating exosomes, which was necessary for the small volumes ideal for the development of a biomarker system, the authors acknowledge the presence of, and measured by NTA, small numbers of other types of Extracellular vesicles (EVs) so wanted to be inclusive in the language used to describe the EV preparation.  The Exo-Check Exosome Antibody arrays (Systems Biosciences, EXORAY200B-4) were used to display that the precipitated extracellular vesicles (EVs) contained exosomes. These arrays have 12 total printed antibody spots, 8 of which are for known exosome markers (CD63, CD81, TSG101, Alix, FLOT1, EpCAM, ICAM1 and ANXA5).

SIMOA analysis should be described in more details.

The authors thank the reviewer for this feedback and have added some additional information at a level of detail accessible from the company’s proprietary information. Single Molecular Array (SIMOA) is an immunoassay-based technique that uses paramagnetic beads to which antibodies for the analytes of interest (e.g., Ab42) are attached. These bead-antibody complexes are used to capture the analytes of interest from the sample. The captured analyte is then measured using a secondary antibody that is tagged with an enzyme that can cleave a fluorescent substrate and provide a signal that reflects the amount of captured analyte. The sensitivity of the assay arises from the ability of the technology to count each individual bead-analyte event for up to 240,000 events in single form with even one captured analyte being sufficient to provide signal of sufficient intensity to be measurable.

When the Aβ 1-42 was added to the isolated EVS, what was its conformation? Was it a monomer, oligomer or matured fibrils? If it was fibrils, then what was the pretreatment done to induce fibrillation. Explain in the methodology section.

The authors thank the reviewer for this feedback. For the “Binding of Ab42 to extracellular vesicles” experiments this was monomeric Ab42. The term monomeric has been added to this method.

Thioflavin T assay was done and it is known that this dye specifically binds to the amyloid structure. How did the authors confirm that the Aβ 1-42 is in its fibrillar form?

The authors have confirmed that Aβ1-42 is fibrillar when treated under these conditions using Electron Microscopy.

The discussion part needs to be improved.

The authors are grateful for this feedback and have added more in the way of discussion to this manuscript.

Explain a little more about the scrambled peptides in the manuscript and how they are taken as non-amyloidogenic proteins.

The scrambled peptides contain all the amino acids found in Ab40 and Ab42 just arranged in a different sequence. Thus they do not form the amyloid structures normally observed for Ab (confirmed by the R-peptide company and our AFM analysis).

General comments:

The English language needs to be corrected in the supplementary files of the manuscript. The figure captions of supplementary files are difficult to understand. Please make it lucid.

The authors thank the reviewer for this feedback and have amended the language used for the supplementary files to make it more understandable.

I recommend major revision.

Comments on the Quality of English Language

The langusge of the supplementary files needs to be improved

Submission Date

23 January 2024

Date of this review

06 Feb 2024 04:36:55

Reviewer 2 Report (New Reviewer)

Comments and Suggestions for Authors

My comments on the manuscript entitled “Specific Binding of Alzheimer’s A peptides to Extracellular 2 Vesicles”

How do extracellular vesicles (EVs) play a role in Alzheimer's disease, and what is their significance in carrying bioactive molecules related to the disease process?

Can you elaborate on the specific findings regarding the interaction between amyloid proteins Aβ42 and Aβ40 with EVs, as discovered through Isothermal Titration Calorimetry (ITC)? How does this interaction contribute to the understanding of Alzheimer's progression?

What is the potential significance of amyloid-EV interaction as a brain-specific biomarker for Alzheimer's disease, and how might it aid in the early detection or monitoring of the disease?

The study mentions that overnight incubation of Aβ with EVs resulted in larger amounts of bound Aβ peptide with a fibrillar structure. How does this observation contribute to our understanding of the progression of Alzheimer's disease, and what implications might it have for future research or diagnostic approaches?

How does the use of Single Molecule Array (SMA) contribute to determining the binding of Aβ42 and Aβ40 amyloid peptides to extracellular vesicles (EVs) in the described experimental setup?

Could you elaborate on the significance of employing scrambled versions of Aβ peptides (Aβ42scr or Aβ40scr) in the incubation process, and what role do they play in confirming the binding specificity of Aβ42 and Aβ40 to EVs?

In what ways do Isothermal Titration Calorimetry (ITC) and Atomic Force Microscopy (AFM) contribute to the analysis of the molecular specificity and thermodynamic properties of the interaction between Aβ peptides and EVs? What insights do these techniques provide?

The text mentions that the results point to Aβ having a more selective interaction with EVs than previously understood. Could you explain how this finding challenges or refines existing understanding and what implications it might have for our comprehension of Aβ-EV interactions in physiological and pathological conditions, particularly in Alzheimer's Disease (AD)?

Can you provide insights into the significance of employing a variety of techniques such as Western Blotting, transcriptomics, proteomics, ELISA, PCR, immunohistochemistry, immunocytochemistry, and electron microscopy in understanding the role of extracellular vesicles (EVs) in Alzheimer's Disease (AD) pathology, specifically focusing on Aβ-EV associations?

The text mentions that while previous studies agree that EVs carry Alzheimer’s Aβ peptides, this study investigates the selectivity of the Aβ-EV association. How does understanding the selectivity of this association contribute to establishing the physiological or pathological relevance of loading Aβ into/onto EVs?

In the context of Isothermal Titration Calorimetry (ITC) results, why does Aβ40 binding exhibit endothermic and saturable characteristics, while the more aggregation-prone Aβ42 binding is peptide-form dependent and more robustly observed when preformed fibrillar Aβ42 (Ab42F) is added to the assay? What implications do these observations have for the association between Aβ and EVs?

The study emphasizes the potential of using Aβ levels on EVs as a dynamic biomarker for monitoring the progression of Alzheimer's Disease. How might the specific and saturable interaction between amyloid and EVs influence potential therapeutic interventions focused on Aβ clearance, especially considering the preference of fibrillar Aβ42 (Aβ42F) for associating with extracellular vesicles?

How do extracellular vesicles selectively and specifically bind to Aβ peptides?

What potential roles do EV-associated amyloid play, ranging from clearance to catalysis of fibrillar Aβ formation?

Could the association of Aβ with extracellular vesicles suggest a role in catalyzing the formation of intracellular fibrillar forms before secretion into the extracellular space?

How might extracellular vesicles released into the extracellular space seed or catalyze the formation of fibrillar forms of Aβ, and what role do the amyloid on their surface or lipid membranes play in this process?

In what ways could Aβ secreted on extracellular vesicles facilitate the uptake of Aβ into naïve cells?

How could the association of Aβ with extracellular vesicles serve as a useful biomarker for monitoring disease progression?

What is the potential significance of Aβ levels on extracellular vesicles as a biomarker in assessing the response to therapeutic interventions?

What ongoing investigations are currently underway to explore the utility of Aβ levels on extracellular vesicles as a biomarker for disease progression and response to therapeutic interventions?

Include some relevant bibliographic studies like PMID: 30605887, & PMID: 30503937

Complete editorial checking will be needed for your manuscript.

Comments on the Quality of English Language

Complete editorial checking will be needed for your manuscript.

Author Response

REVIEWER 2:

Comments and Suggestions for Authors

The authors thank the reviewer for their excitement and interest in this work.

My comments on the manuscript entitled “Specific Binding of Alzheimer’s Ab peptides to Extracellular 2 Vesicles”

How do extracellular vesicles (EVs) play a role in Alzheimer's disease, and what is their significance in carrying bioactive molecules related to the disease process? What is the potential significance of amyloid-EV interaction as a brain-specific biomarker for Alzheimer's disease, and how might it aid in the early detection or monitoring of the disease? Can you elaborate on the specific findings regarding the interaction between amyloid proteins Aβ42 and Aβ40 with EVs, as discovered through Isothermal Titration Calorimetry (ITC)? How does this interaction contribute to the understanding of Alzheimer's progression?

The authors thank the reviewer for their questions which are all excellent. While there is beginning to become an agreement that exosomes carry Ab there is a wide-ranging view on whether the purpose of this association is toxic or beneficial the conclusions of which are dependent on whether the authors of these works consider Ab in its fibrillar form to be more toxic or protective. The questions that are being pursued can be summarized as attempts to understand if EVs clear or spread toxic peptides and pathology? Are EVs binding Ab to reduce the formation of fibrillar forms? Do EVs bind monomers and lead to reductions in the time it takes to form fibrils and/or do they have a preference for one form over another. Our ITC experiments point to a preference for binding to fibrillar forms of Ab42. Questions that still abound are when this Ab-EV binding occurs is this while the EVs are still in the cell (before the EV is released into the Extracellular space) or after it is released? Since EVs can easily traverse membranes and enter cells and cross the Blood Brain Barrier there are many interesting and important investigations that need to be made now that we are starting to understand that amyloid is bound to exosomes. These points have all been added to the discussion in the manuscript.

The study mentions that overnight incubation of Aβ with EVs resulted in larger amounts of bound Aβ peptide with a fibrillar structure. How does this observation contribute to our understanding of the progression of Alzheimer's disease, and what implications might it have for future research or diagnostic approaches?

The authors thank the reviewer for their question. In the Ab/amyloid field controversy still abounds as to the most toxic form of Ab. Research points to many potentials. The monomeric/oligomeric forms may be more toxic due to their mobility and ability to be secreted and cross membranes in a less impeded manner. Others are of the opinion that the fibrillar forms are more toxic and need to be cleared. It is known that if we incubate Ab it will form amyloid as it is aggregation prone. So are the exosomes gathering Ab and clearing it or spreading it pathologically? This is an area of much interest to us and others.

How does the use of Single Molecule Array (SMA) contribute to determining the binding of Aβ42 and Aβ40 amyloid peptides to extracellular vesicles (EVs) in the described experimental setup?

The authors thank the reviewer for their question. SIMOA has sensitivity in the femtomole range allowing us to measure the amounts of Ab40 and Ab42 that have bound to the exosomes. This allowed us to accurately quantify the amount of Ab that was present on/in the exosome post incubation with those peptides. This has now been elaborated on in the Materials and Methods.

Could you elaborate on the significance of employing scrambled versions of Aβ peptides (Aβ42scr or Aβ40scr) in the incubation process, and what role do they play in confirming the binding specificity of Aβ42 and Aβ40 to EVs?

The authors thank the reviewer for their question. The scrambled versions were used to show that the sequence of the amino acids of Ab is important for its binding as no Ab40 or Ab42 signal could be detected when the amino acid sequence of Ab40/Ab42 is rearranged. While these standardized assays do not allow us to detect the scrambled versions themselves we can use these findings to point to the lack of ability of the scrambled versions to act as mimics of Ab40 or Ab42.

In what ways do Isothermal Titration Calorimetry (ITC) and Atomic Force Microscopy (AFM) contribute to the analysis of the molecular specificity and thermodynamic properties of the interaction between Aβ peptides and EVs? What insights do these techniques provide? The text mentions that the results point to Aβ having a more selective interaction with EVs than previously understood. Could you explain how this finding challenges or refines existing understanding and what implications it might have for our comprehension of Aβ-EV interactions in physiological and pathological conditions, particularly in Alzheimer's Disease (AD)?

The authors thank the reviewer for their questions. While AFM has many strengths and uses, in this work we used it to assess the starting forms of peptides used for these experiments and also the forms that resulted post incubation of the Ab peptides with EVs. The fact that overnight incubations of EVs with Ab showed that it was present on EVs and that longer incubations resulted in more fibrillar forms confirming the aggregation capabilities of Ab but also implying/pointing to the possibility that EVs may form a scaffold upon which the formation of fibrillar forms may occur. This assembly of fibrillar Ab/or its preferential binding to EVs when present can then point to the possibility that EVs may seed pathology using this attached Ab/amyloid and/or clear it from the cell/body. Since EVs have been reported to have Ab/amyloid on/in them we wanted to determine if this was a specific association and not just due to the hydrophobic coronas binding to this hydrophobic peptide/amyloid in a non-specific manner. The ITC points to fibrillar forms being capable of interacting with and binding to the EVs (maybe through the corona/lipid bilayer) in a specific and saturable manner that is energy requiring. Given the energy input required if this is occurring in a cell/in the body then this would point to a role for this association given the body does not use energy in a wasteful manner.

Can you provide insights into the significance of employing a variety of techniques such as Western Blotting, transcriptomics, proteomics, ELISA, PCR, immunohistochemistry, immunocytochemistry, and electron microscopy in understanding the role of extracellular vesicles (EVs) in Alzheimer's Disease (AD) pathology, specifically focusing on Aβ-EV associations?

The authors thank the reviewer for their question. We wanted to ensure that what we were isolating and studying in this work was EVs due to at least in large part EVs that were exosomes. Examining the expression of certain proteins (tetraspnnins) on their surface as well as their size and composition allows us to more definitively say that we are working with EVs and not just pure lipoproteins (lipoproteins can/are associated with EVs) or cell debris. We hence wanted to characterize these EVs as EVs using the MISEV guidelines for proving that we had EVs. My understanding is that an updated version of these guidelines is on its way to press but in this work we followed the 2018 version.

The text mentions that while previous studies agree that EVs carry Alzheimer’s Aβ peptides, this study investigates the selectivity of the Aβ-EV association. How does understanding the selectivity of this association contribute to establishing the physiological or pathological relevance of loading Aβ into/onto EVs?

This assembly of fibrillar Ab/or its preferential binding when present to EVs can then point to the possibility that EVs may seed pathology using this attached Ab/amyloid and/or clear it from the cell/body. Since EVs have been reported to have Ab/amyloid on/in them we wanted to determine if this was a selective association and not just due to the hydrophobic coronas binding to this hydrophobic peptide/amyloid in a non-selective manner. The ITC points to this being a more selective and saturable interaction that is energy requiring and hence possibly deliberate and important for cellular physiology and/or pathology.

In the context of Isothermal Titration Calorimetry (ITC) results, why does Aβ40 binding exhibit endothermic and saturable characteristics, while the more aggregation-prone Aβ42 binding is peptide-form dependent and more robustly observed when preformed fibrillar Aβ42 (Ab42F) is added to the assay? What implications do these observations have for the association between Aβ and EVs?

The authors thank the reviewer for their questions. The ITC points to fibrillar forms being capable of interacting with and binding to the EVs (maybe through the corona/lipid bilayer) in a selective and saturable manner that is energy requiring. Given the energy input required if this is occurring in a cell/in the body then this would point to an important role for this association given the body does not use energy in a wasteful manner.

Comparing the surface area of the exosomes against the concentration of Ab if there are ~500k Ab per EV based on EV molarity and amount of Ab added this would lead to an Ab monomer binding every 3 angstroms.  Given this is probably not physically possible, the most reasonable explanation is that Ab oligomerizes before binding and this interaction is selective and important.

The study emphasizes the potential of using Aβ levels on EVs as a dynamic biomarker for monitoring the progression of Alzheimer's Disease. How might the specific and saturable interaction between amyloid and EVs influence potential therapeutic interventions focused on Aβ clearance, especially considering the preference of fibrillar Aβ42 (Aβ42F) for associating with extracellular vesicles? How could the association of Aβ with extracellular vesicles serve as a useful biomarker for monitoring disease progression? What is the potential significance of Aβ levels on extracellular vesicles as a biomarker in assessing the response to therapeutic interventions? What ongoing investigations are currently underway to explore the utility of Aβ levels on extracellular vesicles as a biomarker for disease progression and response to therapeutic interventions?

The authors thank the reviewer for their questions. We have expanded the Discussion and Conclusion in the revised manuscript (attached in responses) to provide oir thoughts on these topics and thank the reviewer for these very interesting questions.

How do extracellular vesicles selectively and specifically bind to Aβ peptides?

The authors thank the reviewer for their question. We have identified some surface EV candidates that we are interrogating and hope to share in another manuscript soon.

What potential roles do EV-associated amyloid play, ranging from clearance to catalysis of fibrillar Aβ formation? Could the association of Aβ with extracellular vesicles suggest a role in catalyzing the formation of intracellular fibrillar forms before secretion into the extracellular space? How might extracellular vesicles released into the extracellular space seed or catalyze the formation of fibrillar forms of Aβ, and what role do the amyloid on their surface or lipid membranes play in this process?

In what ways could Aβ secreted on extracellular vesicles facilitate the uptake of Aβ into naïve cells?

The authors thank the reviewer for their question. EVs can easily cross lipid membranes and we and others have data that they can easily enter cells so if Ab/amyloid is selectively and stably attached we could assume that this Ab/amyloid could then also be brought into a cell in this manner.

Include some relevant bibliographic studies like PMID: 30605887 & PMID: 30503937

The authors thank the reviewer for these articles that will be included in the citations and essence of this work.

Complete editorial checking will be needed for your manuscript. Comments on the Quality of English Language Complete editorial checking will be needed for your manuscript.

The authors thank the reviewer for making sure this manuscript is of the best quality possible and appreciate all the careful checking and editing to make it so.

Submission Date 23 January 2024

Date of this review

31 Jan 2024 08:07:55

Reviewer 3 Report (New Reviewer)

Comments and Suggestions for Authors

I thank the authors for their work on the binding of extracellular vesicles to Aβ peptides. The authors make interesting findings about the association of Aβ and EVs and the possible role of EVs in Aβ clearance.

However, the article needs serious revision and cannot be accepted for publication in its present form.

It is not clear from what biological fluid the EVs were derived. Have exosomes from Alzheimer's patients been used? Or were they from healthy patients? The results mention a control group, but I couldn't figure out what the control group was.

Why were Aβ42, Aβ40 and scrambled peptides chosen? What are scrambled peptides?

The text of the article needs significant revision. In many places it is not possible to understand what is being said. It is necessary to refine the Figures and their captions so that the captions reflect the essence of the Figure. Each drawing must indicate a control.

Specific Comments:

1. Line 213. Please add information about receiving EVs. The control and experimental groups should be mentioned.

2. Fig. 1a, 1c, 3 are small, please increase the image quality.

3. Please move the table. 1 and lines 168-177 in the “results” chapter.

4. Lines 217-226. Please describe Figure 1 (a-d) in more detail. For example, indicate the hydrodynamic radii range of your EVs.

Lines 334-335. “Second, the expression of surface molecules on the EVs that bind to amyloid may need to be increased by the cells releasing the EVs (which could occur in compensation if more EVs are not released).” This assertion is unsubstantiated and raises doubts.

Author Response

REVIEWER 3:

Comments and Suggestions for Authors

I thank the authors for their work on the binding of extracellular vesicles to Aβ peptides. The authors make interesting findings about the association of Aβ and EVs and the possible role of EVs in Aβ clearance.

The authors thank Reviewer 3 for their kind feedback.

However, the article needs serious revision and cannot be accepted for publication in its present form.

It is not clear from what biological fluid the EVs were derived. Have exosomes from Alzheimer's patients been used? Or were they from healthy patients? The results mention a control group, but I couldn't figure out what the control group was.

The authors thank the reviewer for sharing this oversight on the part of the authors. The EVs were isolated from human plasma from healthy subjects (referred to as Controls). This has now been shared in the Materials and Methods section.

Why were Aβ42, Aβ40 and scrambled peptides chosen? What are scrambled peptides?

The scrambled peptides contain all the amino acids found in Ab40 and Ab42 just arranged in a different sequence. Thus they do not form the amyloid structures normally observed for Ab (confirmed by the R-peptide company and our AFM analysis) and were used to show that the sequence of Ab is important for binding to exosomes.

The text of the article needs significant revision. In many places it is not possible to understand what is being said.

The authors thank the reviewer for this feedback and the manuscript has been revised to address any areas that were not understandable. We are grateful for all the feedback that has made this a better manuscript.

It is necessary to refine the Figures and their captions so that the captions reflect the essence of the Figure. Each drawing must indicate a control.

Specific Comments:

  1. Line 213. Please add information about receiving EVs. The control and experimental groups should be mentioned.

The authors thank the reviewer for their feedback and wanted to clarify that Control refers to human plasma from healthy subjects. For clarity we have explained this in the Materials and Methods section.

  1. 1a, 1c, 3 are small, please increase the image quality.

The authors thank the reviewer for their feedback. The image has been increased in size and resolution.

  1. Please move the table. 1 and lines 168-177 in the “results” chapter.

The authors thank the reviewer for their feedback. This table and the corresponding text will be moved into the results section.

  1. Lines 217-226. Please describe Figure 1 (a-d) in more detail. For example, indicate the hydrodynamic radii range of your EVs.

The authors thank the reviewer for their feedback. More details have now been added to the legend for Figure 1.

Lines 334-335. “Second, the expression of surface molecules on the EVs that bind to amyloid may need to be increased by the cells releasing the EVs (which could occur in compensation if more EVs are not released).” This assertion is unsubstantiated and raises doubts.

 The authors thank the reviewer for their feedback. We have removed the assertion as it is based on only preliminary evidence that we are still investigating.

Submission Date

23 January 2024

Date of this review

31 Jan 2024 07:34:13

Reviewer 4 Report (Previous Reviewer 1)

Comments and Suggestions for Authors

The Authors addressed my concerns

Author Response

REVIEWER 4:

Comments and Suggestions for Authors

The Authors addressed my concerns.

The authors are grateful to the reviewer for their feedback and time.

Submission Date

23 January 2024

Date of this review

29 Jan 2024 14:06:42

Reviewer 5 Report (New Reviewer)

Comments and Suggestions for Authors

In this interesting work, the authors investigated the interaction between extracellular vesicles and Abeta peptide in its monomeric and fibrillar forms. The results are very interesting and suggest how extracellular vesicles may have both a physiological role in amyloid clearance and a pathological role as a marker of deposition of the fibrillar form. Microvesicles are very important components in cellular communication even at a distance. They may carry up- and downregulated micro RNAs that discriminate various forms of neurodegenerative diseases, including AD (Sproviero D et al.  Extracellular Vesicles Derived From Plasma of Patients With Neurodegenerative Disease Have Common Transcriptomic Profiling.  doi: 10.3389/fnagi.2022.785741), or be transporters of extracellular proteins such as Abeta. I congratulate the authors for this excellent and also well-written paper, but recommend a better discussion of the role of microvesicles as suggested here, both in the introduction and in the discussion, to provide the reader with an appropriate framework to facilitate reading and understanding of the paper.

Author Response

REVIEWER 5:

Comments and Suggestions for Authors

In this interesting work, the authors investigated the interaction between extracellular vesicles and Abeta peptide in its monomeric and fibrillar forms. The results are very interesting and suggest how extracellular vesicles may have both a physiological role in amyloid clearance and a pathological role as a marker of deposition of the fibrillar form.

The authors are very grateful for this kind feedback.

Microvesicles are very important components in cellular communication even at a distance. They may carry up- and downregulated micro RNAs that discriminate various forms of neurodegenerative diseases, including AD (Sproviero D et al.  Extracellular Vesicles Derived From Plasma of Patients With Neurodegenerative Disease Have Common Transcriptomic Profiling.  doi: 10.3389/fnagi.2022.785741), or be transporters of extracellular proteins such as Abeta. I congratulate the authors for this excellent and also well-written paper,

The authors are grateful for the very kind feedback and for the reviewer sharing this important manuscript with us. We have included this manuscript in our citations.

but recommend a better discussion of the role of microvesicles as suggested here, both in the introduction and in the discussion, to provide the reader with an appropriate framework to facilitate reading and understanding of the paper.

The authors are very grateful for this feedback and have expanded on the description, role and importance of microvesicles in the manuscript.

Submission Date

23 January 2024

Date of this review

12 Feb 2024 12:03:47

Round 2

Reviewer 1 Report (New Reviewer)

Comments and Suggestions for Authors

The authors have made the changes as suggested by the reviewers. I have only a small concern that can be addressed during teh galley proof correction. please use SI units to represent the physical quantities. for ex: the unit of time is 'min' not 'mins', similarly for the volumes the units are 'ml or ul' not 'uls'. make corrections in the final galley. I accept the amnuscript in its present form.

Reviewer 2 Report (New Reviewer)

Comments and Suggestions for Authors

The manuscript has been revised as per my suggestions. 

Reviewer 3 Report (New Reviewer)

Comments and Suggestions for Authors

The new additions to the manuscript made a big difference. The quality of the paper had improved, and all my questions were addressed. No more comments.

This manuscript is a resubmission of an earlier submission. The following is a list of the peer review reports and author responses from that submission.

Round 1

Reviewer 1 Report

Comments and Suggestions for Authors

The manuscript has been improved from the previous submission. However, there are still some typos that need to be corrected. For instance: at page 3, "nm" must be used instead of "nms"; x-axis of figure 1 B and C are not homogeneously labelled.

Author Response

Reviewer 1:

The manuscript has been improved from the previous submission.

The authors thank Reviewer 1 for this helpful feedback.

However, there are still some typos that need to be corrected. For instance: at page 3, "nm" must be used instead of "nms"; x-axis of figure 1 B and C are not homogeneously labelled.

We have addressed the typos that were shared and thank Reviewer 1 for their feedback and insights. For Fig 1 B and C these represent exosomes that were incubated with Ab overnight and thus are different from 1A that reflects the Ab42, Ab40 and Tau that is present on control exosomes before incubation.

Reviewer 2 Report

Comments and Suggestions for Authors

The quality of ITC data in this paper is terrible, unacceptable for publishing and the data is not interpreted clearly by the authors. More controls should be done for ITC experiments as suggested in my previous review . 

I do not recommend it for publishing at current form.

Author Response

Reviewer 2:

The quality of ITC data in this paper is terrible, unacceptable for publishing and the data is not interpreted clearly by the authors. More controls should be done for ITC experiments as suggested in my previous review . I do not recommend it for publishing at current form.

 We appreciate the feedback provided by reviewer 2. While the specific nature of the criticism was not detailed, we recognize the importance of addressing potential ambiguities or shortcomings in our work. As a result, we sought the expertise of an additional Biophysics specialist (Dr. Robb Welty whom we will add to the authorship on the paper).

Upon consultation, it was posited that the main critiques might arise from:

  1. The manner in which the data was originally presented, and
  2. The initial interpretation of the data.

Originally, our data presentation adhered to conventional Isothermal Titration Calorimetry (ITC) norms, namely the upper plot of the ITC raw data thermograph accompanied by a binding isotherm below. However, we acknowledge that our experiments diverged from traditional ones. This is not to question the validity of our experiments but to emphasize that conventional presentation methods may not capture the essence of our findings, especially since our focus isn't on traditional binding models like 1:1 or 1:2 sequential binding.

Our core observation is the endothermic reaction occurring when exosomes are mixed with Ab40 and Ab42, a deviation from the typical behavior observed in Ab fibrillization. We hypothesize that this is attributable to a reorganization of the exosomal membrane or to an interaction of Ab with proteins that form part of/associate with the exosomal membrane or surface when exosomes are secreted into plasma. The precise nature of this reorganization or a full interrogation of the proteins on the exosome surface that may interact with Ab was not the focus of this work , but its occurrence is evident and will be interrogated in further studies. We believe we are the first to report this phenomenon concerning Ab and exosomes.

In light of these insights, we have revisited our manuscript to present and interpret our findings in a manner that we believe addresses the potential concerns. We hope that this revised version provides clarity and underscores the novelty of our work. If there remain specific areas of contention, we would greatly appreciate detailed feedback to facilitate further improvements.

Reviewer 3 Report

Comments and Suggestions for Authors

  This current article entitled “Binding of Alzheimer’s Aβ peptides to Exosomes” by Coughlan et al was devoted to determining the interaction between Aβ peptides and exosomes in a specific manner. They have applied Isothermal Titration Calorimetry to measure the binding of Aβ42 and Aβ40 to exosomes, discovering that both peptides bound in a saturable and endothermic manner. The authors mainly focus on determining the interaction between Aβ peptides and exosomes using biophysical technique which I believe beyond the scope of “The International Journal of Molecular Sciences.” I think authors will find a more specific biophysical journal that aligns with their work.  

The following points can improve their work for submission to another journal:

1.      Add some characterization techniques (TEM, zeta size, western blot, etc.) of the isolated exosome.

2.      It is mandatory to mention how much/many exosomes they were able to isolate from the plasma and the final number/conc. of exosomes used for their biophysical assay.

3. The ITC measurement showed that the interaction between the exosome and the Aβ peptides exhibited an endothermic exotherm while scrambled peptides (Aβscr) exhibited negligible exothermic isotherms. Additionally, to explain this property they need to perform zeta-potential titration by following the articles: 10.1039/C5MD00260E, 10.1016/j.bbrep.2015.05.007.

4.      Materials and Methods section: “The exosomes were then reprecipitated from this solution, lysed in 25 mls of 0.05M Glycine and centrifuged @4,000g for 10mins.” Usually, exosome precipitation needs to be done with high-speed centrifugation techniques unless the precipitation is done using some specific polymers. The author should recheck the used RCF value (@4000g) to precipitate the exosome from a buffer solution.

Author Response

Responses attached as a .docx as they have images and tables in them

Round 2

Reviewer 3 Report

Comments and Suggestions for Authors

This current article entitled “Binding of Alzheimer’s Aβ peptides to Exosomes” by Coughlan et al was devoted to determining the interaction between Aβ peptides and exosomes in a specific manner. They have applied Isothermal Titration Calorimetry to measure the binding of Aβ42 and Aβ40 to exosomes, discovering that both peptides bound in a saturable and endothermic manner. The authors mainly focus on determining the interaction between Aβ peptides and exosomes using biophysical technique which I believe beyond the scope of “The International Journal of Molecular Sciences.” I think authors will find a more specific biophysical journal that aligns with their work.